# Effects of Sodium Chloride on the Physical and Oxidative Stability of Filled Hydrogel Particles Fabricated with Phase Separation Behavior

**DOI:** 10.3390/foods10051027

**Published:** 2021-05-09

**Authors:** Chuanai Cao, Xin Li, Yongchao Yin, Baohua Kong, Fangda Sun, Qian Liu

**Affiliations:** 1College of Food Science, Northeast Agricultural University, Harbin 150030, China; cca930504@hotmail.com (C.C.); yingyongchao233@hotmail.com (Y.Y.); kongbh63@hotmail.com (B.K.); sunfangda@neau.edu.cn (F.S.); 2Sharable Platform of Large-Scale Instruments & Equipments, Northeast Agricultural University, Harbin 150030, China; lixin810910@hotmail.com; 3Heilongjiang Green Food Science & Research Institute, Harbin 150028, China

**Keywords:** filled hydrogel, sodium chloride (NaCl), physical stability, oxidative stability, electrostatic interaction

## Abstract

The objective of this study was to investigate the influence of sodium chloride (NaCl) concentration (0–500 mM) on the physical and oxidative stabilities of filled hydrogel that were stabilized using heat-denatured whey protein concentrate and high methoxy pectin. Our results showed that with an increase in NaCl concentration, the particle sizes, zeta-potentials, and interfacial layer thickness of filled hydrogels significantly increased and the lightness and whiteness gradually decreased (*p* < 0.05). Moreover, rheological characterization revealed that the apparent viscosity and viscoelastic behavior gradually decreased at higher NaCl concentration, which was mainly ascribed to the influence of NaCl on the electrostatic repulsion between droplets, thereby adversely impacting the physical stability of filled hydrogels. Furthermore, the result of cryo-scanning electron microscopy also verified the abovementioned results. Notably, higher NaCl concentration significantly promoted the oxidation of lipids and proteins (*p* < 0.05), thereby decreasing the oxidative stabilities of filled hydrogels. Our results indicated that filled hydrogels prepared under different ionic strength conditions can provide the theoretical basis for their future application in emulsion-based foods.

## 1. Introduction

In recent years, emulsion-based delivery systems such as oil-in-water emulsions (O/W), multilayer (O_M_/W), water-in-oil-in-water (W/O/W), and oil-in-water-in-water (O/W/W) emulsions, are well-suited to encapsulate or incorporate bioactive lipids or compounds such as lipid-soluble vitamins, ω-3 fatty acids, curcumin, and carotenoids into aqueous foods or beverages [1,2]. As a novel emulsion-based delivery system, filled hydrogels have considerable potential for specific applications [3,4]. The filled hydrogels are typical O/W/W emulsions assembled by mixing an O/W emulsion and a water-in-water (W/W) emulsion according to the biopolymer phase separation method, wherein oil droplets are dispersed within a watery phase that is dispersed within another watery phase [5]. Compared with other conventional emulsions, filled hydrogels have the potential to provide benefits such as improved stability, effective controlled release of bioactive compounds, and improved overall flavor intensity for the reduced fat emulsions system [1,2,6]. Notably, emulsion-based systems experience various environmental conditions such as changes in ionic strength, pH, temperature, and mechanical mixing speed during manufacture of commercial food or medicinal products [7]. Moreover, emulsion systems may be exposed to various pH levels and ionic strengths at different phases of gastrointestinal digestion after consumption [8,9]. Therefore, it is necessary to design an emulsion system that performs stably under the influence of various factors.

Emulsion systems are particularly sensitive to ionic strength that is known to influence the electrostatic interaction between droplets, thereby inducing protein aggregation and changing the conformational structures of proteins, which might cause instability in protein-stabilized O/W emulsion [10,11]. Tcholakova et al. [12] reported that ionic strength could induce the flocculation or aggregation of droplets and decrease the physical stability of O/W emulsions. Moreover, an increase in ionic strength weakened the attractive electrostatic interactions between two biopolymer molecules at the droplet surface, thereby promoting bridging flocculation of an emulsion stabilized by protein/polysaccharide complexes [13,14]. Furthermore, ionic strengths have been shown to influence the lipid oxidation of food emulsions, thereby affecting the shelf life. Shao et al. [15] showed that increasing ionic strength produced more negative charges and improved chelation capacity of lecithin, resulting in iron enrichment near the droplet surface and enhanced lipid oxidation of soy protein isolate-stabilized emulsions. Although the effects of ionic strength on the physicochemical characteristics of common emulsions have been widely studied, there are limited studies about the effects of ionic strengths on physical and oxidative stabilities of filled hydrogels.

In our previous study, we successfully prepared filled hydrogels by combining an O/W emulsion with the upper and lower phases generated by phase separation behavior of heat-denatured whey protein concentrate (HWPC) and anionic high methoxy pectin (HMP) under acidic conditions (pH 5.0) [16]. Whey protein concentrate is obtained by removing non-protein components from pasteurized whey. After heating over denaturation temperatures, whey protein will expose more hydrophobic groups and sulfhydryl groups buried in the molecule due to the partial unfolding of the globular protein. Moreover, as a natural anionic polysaccharide, HMP has been widely used in the food industry because of good gelling and thickening properties. Therefore, the change in protein structure and electrostatic interaction between HWPC and HMP at neutral pH would be attributed to the occurrence of phase separation behavior and the formation of filled hydrogels. The results demonstrated that the optimum stability of filled hydrogels was at a HWPC/HMP mass ratio of 3:1 during phase separation. The main objective of this study was to investigate the influence of sodium chloride (NaCl) concentration (0–500 mM) on the physical (particle size, zeta-potential, rheological and optical properties, as well as microscopic structure) and oxidative stabilities (lipid and protein oxidation) of filled hydrogel particles. These results could contribute to the design and manufacture of novel delivery systems for specific applications in a range of functional food products.

## 2. Materials and Methods

### 2.1. Materials

Whey protein concentrate (WPC) containing 80% protein based on total weight was obtained from Milkyway Trade Co., Ltd. (Beijing, China). High methoxy pectin (HMP) was purchased from Yuanye Biological Technology Co., Ltd. (Shanghai, China) with the degree of esterification at 65%. Canola oil was obtained from Fengzhong Technology Co., Ltd. (Tianjin, China). Polystyrene latex aqueous suspension (PLAS) was procured from Murray Biotechnology Co., Ltd. (Shanghai, China). All other chemicals and reagents were of analytical grade.

### 2.2. Preparation of Phase Separation

Before preparation, the pre-prepared solutions of HWPC and HMP (3%, *w*/*w*, dry weight) were obtained following the procedure described in our previous study [16]. The HWPC solution was then mixed with HMP solution at a mass ratio of 3:1 and the pH of the mixture was maintained at 7.0. Next, the mixture was stirred continuously at 1000 rpm for 30 min and then centrifuged at 10,000× *g* for 2 h at 4 °C using GL-21 M centrifuge (Xiangyi Ltd., Changsha, China). Finally, the upper and lower phases were collected separately.

### 2.3. O/W Emulsion Formation

Emulsions were prepared by mixing 20% canola oil with 80% WPC solution (20 mg/mL) and homogenized at 15,000 rpm for 2 min and then further homogenized in a high-pressure homogenizer (ATS Engineer, Inc., Shanghai, China) at 30 MPa for two cycles.

### 2.4. Filled Hydrogel Preparation

The preparation procedures of filled hydrogels at different NaCl concentrations are shown in Figure 1. The filled hydrogel particles were prepared according to the procedure described by Matalanis et al. [17] with a few modifications. First, the O/W emulsion was mixed with lower and upper phases at mass ratio of 1:1:18, and then stirred for 45 min at 1000 rpm at 25°C. Then, the pH of mixtures was adjusted to 5.0, and NaCl solution (4 M, pH 5.0) was added to aliquots making the final NaCl concentrations of 0, 100, 200, 300, 400, and 500 mM in the respective filled hydrogel systems. Sodium azide (0.02%, *w*/*w*) was added to inhibit microbial growth in the filled hydrogels.

### 2.5. Measurements of Droplet Size, Size Distribution, and Zeta-Potential

The determination of particle size and distribution, as well as zeta-potential of each filled hydrogel was analyzed by using the Malvern 2000 laser particle size analyzer (Malvern instruments Ltd., Worcestershire, UK) and the Nano ZS dynamic light scattering instrument (Malvern instruments Ltd., Worcestershire, UK), respectively.

### 2.6. Rheological Behavior

Two types of oscillatory measurements, steady shear and frequency sweep tests, were conducted using a controlled stress rheometer (DHR-1, TA Instruments, NewCastle, DE, USA) equipped with serrated plate-plate geometry (40 mm diameters) and a gap between parallel plates of 1 mm. The sample was placed directly on the bottom plate and the rim was sealed off with a low viscosity paraffin oil before measurement.

#### 2.6.1. Steady Shear Tests

Apparent viscosities curves were constructed through continuous steady shear tests varying the shear rate from 0.1 to 100 s^−1^. The obtained curves were suitable for the Cross model that was used to evaluate the dependence relationship between shear rate and apparent viscosity as follows:(1)$$\frac{\eta -\eta_{\infty }}{\eta_{0}-\eta_{\infty }}\;=\frac{1}{1+{\left(\lambda \gamma \right)}^{m}}$$
where *η* represents apparent viscosity (Pa·s); *η*_0_ represents zero-shear viscosity (Pa·s) at low shear rate; *η*_∞_ represents infinite-shear viscosity (Pa·s) at the high shear rate; *γ* represents the shear rate (s^−1^); *λ* represents the time constant (s); *m* represents the dimensionless exponent.

#### 2.6.2. Frequency Sweep Tests

The viscoelastic properties were characterized within linear viscoelastic range. The frequency sweep tests were measured between the range of 1–100 Hz with the strain value set at 1%. The frequency dependencies of storage modulus (*G*′) and loss modulus (*G*″) were approximated using the power law model:*log G*′ = *log a*′ + *b*′ *log f*(2)
*log G*″ = *log a*″ + *b*″ *log f*(3)
where, *f*, *a* (*a*′ and *a*″), and *b (b*′ and *b*″) are the frequency (Hz), the model parameters (Pa/sⁿ), and the dimensionless frequency indices, respectively.

### 2.7. Interfacial Layer Thickness

According to the description of Wong et al. [18], 0.5 g of 0.1% PLAS was mixed with 2.0 g of 2.0% WPC and then incorporated into 45.0 g of the upper phase and 2.5 g of the lower phase. The mixtures were then stirred continuously at 1000 rpm for 30 min and the pH was adjusted to 5.0. Finally, NaCl solution (4 M, 10 mM phosphate buffer, pH 5.0) was added to acquire a series of mixtures with NaCl concentrations of 0, 100, 200, 300, 400, and 500 mM in filled hydrogels. The average particle diameter was measured for each mixture at 25 °C.

### 2.8. Color Measurement

Before testing, the color meter was standardized using white and black calibration plates. The color (*L**, *a**, *b**) of each sample was measured based on the procedure described in our previous study [19]. Whiteness was calculated using Equation (4) as follows:(4)$$\mathrm{Whiteness}=100-\sqrt{{\left(100-L^{*}\right)}^{2\;}+a^{*}{}^{2}+b^{*}{}^{2}}$$

### 2.9. Cryo-Scanning Electron Microscopy (Cryo-SEM)

Micromorphology of each sample was observed with scanned images magnified 10,000× using a Cryo-SEM equipped with a cryo-transfer system (S-3400N, Hitachi, Japan) according to the methods of our previous study [19].

### 2.10. Lipid Oxidation Measurements

#### 2.10.1. Conjugated Dienes (CD)

The CD value was evaluated using the method described in the study of Li et al. [20]. In brief, 0.1 mL of each filled hydrogel was mixed with 1.5 mL isooctane-isopropanol (3:1, *v*/*v*) in 10 mL centrifuge tube and then vortexed three times for 10 s using a vortex mixer (3030A, Scientific Industries, INC., Bohemia). Each mixture was then centrifuged at 550× *g* for 5 min and the organic solvent phase was collected. Finally, 0.1 mL organic solvent phase was incorporated with 4.8 mL isooctane and the absorbance value of the mixture was measured at a wavelength of 234 nm.

#### 2.10.2. Thiobarbituric Acid-Reactive Substances (TBARS)

The TBARS measurements were conducted according to the method described in our previous study [16]. The TBARS values were calculated based on the standard curve generated using 1,1,3,3-tetraethoxypropane.

### 2.11. Protein Oxidation Measurements

The degree of protein oxidation was evaluated according to the procedure described by Cao et al. [19]. In order to determine the natural tryptophan fluorescence, the excitation wavelength was set at 283 nm and the fluorescence was scanned from 300 to 400 nm. The tryptophan fluorescence of protein oxidation products (FP) was determined by recording the fluorescence from 400 to 500 nm at the excitation wavelength of 350 nm.

### 2.12. Statistical Analyses

Three batches (replicates) of each sample were prepared independently and indices of each batch were measured three times as technical repetition. Data were displayed as the mean ± standard deviations (SD). The analysis of variance (ANOVA) was performed using the Tukey procedures to measure the significance of the main effects (*p* < 0.05). Correlation analyses were performed for all instrumental variables (particle size, zeta-potential, lipid oxidation, and protein oxidation) using the R-3.5.3 software (version 3.5.3, Tsinghua University, Beijing, China).

## 3. Results and Discussion

### 3.1. Particle Size and Distribution

An aim of this experiment was to examine the influence of NaCl concentration on the physical stability of filled hydrogels. All freshly prepared filled hydrogels showed a uniform and white appearance in Figure 2A, which was supported the whiteness results in color analysis. Moreover, the particle size distributions of freshly prepared filled hydrogels at different NaCl concentrations are shown in Figure 2C. The large particle peak in the particle size distribution of filled hydrogels shifted toward the right with the increase in NaCl level increased from 0 to 500 mM, suggesting an increase in particle size. Meanwhile, Table 1 shows that as the NaCl level increased from 0 to 500 mM, the volume averaged diameter (*D_4,3_*) and the surface-averaged diameter (*D_3,2_*) of the freshly prepared filled hydrogel increased (*p* < 0.05). For instance, when compared with filled hydrogel without NaCl, the *D_4,3_* of filled hydrogel increased by 1.63, 12.75, 22.50, 25.85, and 32.12% for NaCl concentrations of 100, 200, 300, 400, and 500 mM, respectively. This phenomenon was mainly attributed to electrostatic screening, ion binding, interfacial rearrangements, or polysaccharide desorption [21,22]. Harnsilawat et al. [23] reported the gradual screening of electrostatic repulsion among droplets gradually increased with increasing NaCl concentration, eventually leading to the coalescence or creaming of droplets. Moreover, the presence of NaCl weakened the electrostatic attraction between the protein and polysaccharide molecules around the droplet surface, which notably promoted polysaccharide molecules to attach to more than one droplet, resulting in the occurrence of bridging flocculation with large average particle sizes [14].

After 10 days, the filled hydrogels showed a stability phenomenon without creaming from 0 to 300 mM (Figure 2B). However, when the NaCl concentration further increased, the filled hydrogels showed the stratification phenomenon that the upper layer was creamy and the layer phase was a transparent continuous phase (Figure 2B), indicating that droplets became aggregated and phase separation phenomenon appeared due to the imbalance between oil droplets and water phase. Moreover, this phenomenon became more and more obvious with increasing NaCl concentration from 400 to 500 mM. This indicated that the electrostatic repulsion between droplets decreased due to the electrostatic screening of NaCl, thereby decreasing the stability of filled hydrogels. In addition, the filled hydrogels stored for 10 days had higher particle sizes (*D_4,3_* and *D_3,2_*) than those of the freshly prepared filled hydrogel (*p* < 0.05), indicating that the stability diminished with an increase in storage period. For filled hydrogels stored for 10 days, the droplet diameters of filled hydrogels increased and the large particle peak in the particle size distribution of filled hydrogels shifted toward the right with an increasing NaCl level (Figure 2D and Table 1), which was in accordance with the results of freshly prepared filled hydrogels. NaCl addition promoted the flocculation of protein on the surface of droplets and changed the structure of the interface, causing a larger and non-uniform distribution of droplets. These consequences confirmed that the addition of NaCl might reduce the physical stability of filled hydrogels during storage.

### 3.2. Zeta-Potential

The intensities of electrostatic interactions between droplets at different NaCl levels were determined from the movement of droplets in zeta-potential. As shown in Figure 3, the freshly filled hydrogel without NaCl was negatively charged with absolute value of more than 30 mV, indicating that the filled hydrogels had good stability because of comparatively strong electrostatic repulsion. Moreover, the zeta-potential of filled hydrogels showed a negative charge between −28.30 and −25.57 mV (absolute values lesser than 30 mV) for a series of NaCl concentrations, suggesting that the system tends to become unstable and easily leads to an aggregation of droplets. Notably, the zeta-potential (absolute value) decreased remarkably with an increase in NaCl concentration (*p* < 0.05); the filled hydrogel had the lowest zeta-potential at an NaCl concentration of 500 mM. The results of zeta-potential are similar to those found by Matalanis et al. [17], who reported that the addition of NaCl could significantly decrease the electrical charge of the emulsion fabricated by egg yolk/*κ*-carrageenan composite aqueous. This could be attributed to the electrostatic charge screening properties and/or ion binding effect of NaCl that obviously enhanced with increasing NaCl concentration [24,25]. Matalanis et al. [17] found that NaCl could screen the electrostatic repulsion between the hydrogels, causing an aggregation of particles and a decrease in charge. Moreover, Wu et al. [26] indicated that when NaCl was added, Na^+^ was preferentially adsorbed onto the negatively charged carboxylic groups (-COO−) on the polysaccharides or proteins and Cl^−^ was adsorbed onto positive amino groups (-NH^3+^) on the proteins, thereby leading to a reduction in effective charge density. In addition, Niu et al. [13] suggested that the electrostatic interactions between ovalbumin-stabilized droplet and gum Arabic would be weakened by the addition of NaCl, leading to the desorption of the gum Arabic molecule with negative charge. Interestingly, an obviously lower absolute zeta-potential value of the filled hydrogels was observed when the storage period was increased from 0 to 10 days (*p* < 0.05), suggesting that the filled hydrogels became unstable because of an imbalance of droplets and the water phase for a prolonged storage period. The zeta-potential of filled hydrogels in storage for 10 days significantly decreased with increasing NaCl concentration, which was in accordance with the results for freshly filled hydrogels.

### 3.3. Apparent Viscosity

Evaluating the flow behavior of filled hydrogel particles was important because it could effectively reflect their practical application in the manufacture of food products. As shown in Figure 4, there was an obviously decrease in apparent viscosity as the shear rate was increased from 0.1 to 100 s^−1^ for all the filled hydrogels, suggesting a typical shear thinning flow behavior. As shown in Table 2, the correlation coefficient (R^2^) for each filled hydrogel was above 0.97, indicating that the flow curve was accurately described by the Cross model. Moreover, the value of dimensionless exponents (*m*) tends to 1, which also confirmed a non-Newtonian shear-thinning behavior. The main reason for this behavior was the destruction of molecular aggregation and arrangement caused by an increasing shear rate; the destruction rate was higher than the reintegration rate, thereby leading to transformational behavior from high viscosity to low viscosity [27,28,29].

In Cross model, zero-shear viscosity (*η*_0_) increased when the NaCl concentration increased from 0 to 200 mM (*p* < 0.05), indicating that the addition of NaCl increases the resistance to flow due to the interaction. Moreover, with an increase in the concentration of NaCl from 0 to 200 mM, the apparent viscosity of filled hydrogels increased. Similar results were observed in the study by Shao et al. [30], who reported that the addition of NaCl could increase the viscosity of hydrocolloids and stabilized the emulsion; thus, the viscosity of the emulsion varied with the shear rate. Griffin et al. [31] found that after the adsorption of xanthan gum onto the surfaces of whey protein-coated droplets, the addition of NaCl promoted the bridge between droplets owing to the electrostatic screening effect that contributed to the association between protein-coated droplets and further increasing the viscosity. In addition, Liu et al. reported that the addition of NaCl increased the viscosity of emulsion, which confirmed that non-adsorbed biopolymer (such as protein or polysaccharide) in emulsion after the addition of NaCl may be enhance the viscosity in aqueous phase, resulting in astricting the movement of droplets with a higher viscosity [32]. However, the apparent viscosity of filled hydrogels slowly decreased when the NaCl concentration further increased from 300 to 500 mM. Meanwhile, *η*_0_ decreased with the increase in NaCl concentration (*p* < 0.05). A similar phenomenon was observed in a previous study by Sriprablom et al. [33], who demonstrated the effect of NaCl concentration on the rheological properties of an emulsion stabilized by whey protein and xanthan gum. The flow behavior of proteins and polysaccharides at different NaCl concentrations may determine the apparent viscosity of filled hydrogels. Cai et al. [27] reported that the structure of xanthan gum transformed from ordered (helix) to disordered (coil) when NaCl was added, thereby changing the flow behavior with low apparent viscosity. Moreover, Hao et al. [34] found that increasing NaCl concentration caused the charge shielding effect and made the polysaccharide molecules shrink, thus expressing a lower viscosity. Notably, the time constant (*λ*) increased and then decreased with increasing NaCl concentration in Table 2, which also verified the aforementioned apparent viscosity phenomenon.

### 3.4. The Viscoelastic Characteristics

The dynamic rheological measurements of *G*′, *G*″, and the phase angle tangents (tan *δ*) for filled hydrogels as a function of NaCl concentration have been described in Figure 5. Both *G*′ and *G*″ gradually increase as the frequency is increased from 0.1 to 100 rad/s, elucidating a frequency-dependent viscoelastic behavior. Moreover, *G*′ was lower than *G*″ and tan *δ* was higher than 1 at frequencies ranging from 0.1 to 10 rad/s for all of the filled hydrogels, indicating that liquid-like viscous behavior dominated. However, *G*′ was crossed over and higher than *G*″ with tan δ lower than 1 when the frequency was increased further, indicating a predominantly solid-like behavior. This result confirms the results of the study by Sriprablom et al. [33], who noted the effects of NaCl concentration on the rheological properties of emulsions containing whey protein isolate and anionic xanthan gum. The addition of NaCl induced an increase in both *G*′ and *G*″ values and a decrease in tan *δ*, suggesting that adding NaCl enhanced the viscoelastic properties of filled hydrogels. Li et al. [35] pointed out that NaCl could decrease the repulsive force among molecules and promote the aggregation, entanglement, and network formation in egg yolk/*κ*-carrageenan emulsion systems. However, further increasing the NaCl concentration led to a decrease in *G*′ and *G*″ and an increase in tan *δ*, which indicated weakening of the emulsion structure. Sriprablom et al. [33] reported that the addition of NaCl resulted in electrostatic screening effects that contributed to the coalescence of droplets with a decrease in the viscoelastic behavior. Moreover, a higher NaCl concentration may produce the electrostatic shielding for the protein and decrease the solubility of the protein, which causes the inhibition of hydrogen bonds and hydrophobic interactions between protein and polysaccharide, leading to the lower *G*′ and higher tan δ [36].

The frequency-dependent behaviors of filled hydrogels with different NaCl concentrations were quantitatively evaluated using the power law model. As shown in Table 3, the correlation coefficient (R^2^) for each filled hydrogel was above 0.95, indicating that the viscoelastic behavior of filled hydrogels accurately described by power law model. The *a*′ value is always significantly lower than the *a*″ value at a given NaCl concentration. Moreover, the a′ and *a*″ values visibly increased and then decreased with an increase in NaCl concentration, which was in accordance with the viscoelastic measurements. Notably, *b*′ and *b*″ values for each filled hydrogel were in the range of 0.66 to 1.22 (*b*′ and *b*″ higher than 0), which represented weak gel behavior. We also observed that *b*′ and *b*″ gradually decreased with an increase in NaCl concentration from 0 to 200 mM (*p* < 0.05), indicating that the addition of NaCl reduced the sensitivity of the filled hydrogels to frequency. However, there was no statistically significant difference as the NaCl concentration increased further (*p* > 0.05).

### 3.5. Color Analysis

It is essential to evaluate the color for filled hydrogels because the appearance of food is important for acceptance from consumers. As shown in Table 4, for filled hydrogel without NaCl, *L** value was 59.86%, which was mainly attributed to light scattering by the fat droplets. Moreover, with an increasing concentration of NaCl, there is an obviously decreasing trend in the *L** values of the filled hydrogels (*p* < 0.05). Similarly, Chung et al. [37] concluded that increasing NaCl concentration progressively decreased the *L** value of hydrogels. This phenomenon was caused by changes in particle dimensions and morphology [38]. Moreover, according to our previous study [16], increasing particle size would result in an increase in the dimensions, weakening the refractive index and the scattering efficiency compared with those of the surrounding aqueous phase. This result corroborated the results of particle size of filled hydrogels with different amounts of NaCl. In addition, the whiteness of filled hydrogels increased with increasing NaCl concentration, which was in accordance with the results of *L** value. It is noted that there were no statistically significant differences for *a** value and *b** value, indicating that the addition NaCl had almost no effect on the redness and yellowness of filled hydrogels.

### 3.6. Interfacial Layer Thickness

Figure 6 shows the interfacial thickness of each filled hydrogel as a function of NaCl concentration. Interfacial thickness tended to gradually increase with increasing NaCl concentration for all filled hydrogels and they reached the highest value at 500 mM NaCl. One reason was that the presence of NaCl enhanced surface charge screening and intermolecular interactions, which promoted protein adsorption and the formation of aggregated flocs at the droplet interface, thereby contributing to the formation of a continuous interfacial film with an increase in the interfacial thickness [39,40]. Another reason was that NaCl concentration weakened the electrostatic attraction between the negative groups on the polysaccharides and the positive groups on the protein-coated droplets, thereby inducing a rearrangement of polysaccharide molecules at the droplet surface with a thicker polysaccharide layer [41]. The measured result was in accordance with the particle size results for filled hydrogels with different NaCl concentrations.

### 3.7. Microscopic Morphology

The microstructure of NaCl-treated filled hydrogels is shown in Figure 7A. The droplets were coated by a molecular layer of whey proteins and polysaccharide, contributing to the stabilization of filled hydrogels. Wijaya et al. reported that the stabilization of emulsion stabilized by complexes was due to the intermixed layer of whey protein and low methoxyl pectin or the combination of an interpolyelectrolyte network that formed by pectin chains linked together with clustered protein [42]. With an increase in NaCl concentration, the lipid droplets size gradually increased, which was supported by the particle size results. Moreover, when the NaCl concentration increased, the surface of droplets had some irregular large aggregates. Taha et al. reported that the increase in electrostatic screening induced by NaCl would enhance the interparticle interaction of protein and increase the protein aggregation at the oil/water interface, thus resulting in the flocculation of droplets in emulsion [39]. Moreover, under relatively higher NaCl concentrations (such as 300, 400 and 500 mM), some irregular large aggregates gradually increased in the continuous phase surrounding the droplets, especially indicated in Figure 7E,F. The addition of NaCl may have led to the electrostatic screening and decreased the electrostatic attraction between the protein and pectin molecules, resulting in the protein molecules moving from the filled hydrogels into the continuous phase [43]. Based on the above discussion, Cryo-SEM analysis suggested that the addition of NaCl could account for the physical stability of filled hydrogels.

### 3.8. Lipid Oxidation

Lipid oxidation is an important factor in evaluating the quality characteristics, which is always accompanied by the formation of primary and secondary oxidation products. The primary oxidation products of filled hydrogels are shown in Figure 8A, CD values markedly increased at 37 °C with an increase in the storage time from 0 to 10 days (*p* < 0.05). For instance, the CD values for the filled hydrogel without NaCl observably increased from 1.73 to 14.04 µM as the storage period increased. This indicated that the lipid oxidation did occur in filled hydrogels. Notably, there was no significant difference between any of the freshly prepared filled hydrogels (*p* > 0.05). Moreover, after incubation for 10 days, when the NaCl concentration was less than 300 mM, there was a slight decrease in CD values but no significant difference compared with the filled hydrogel without NaCl (*p* > 0.05). The formation of a secondary thick layer around the droplets may have contributed to improving the oxidative stability of the emulsion [31]. Our aforementioned interfacial layer thickness results indicate that there was no significant difference in filled hydrogels when the NaCl concentration was less than 300 mM; therefore, increasing NaCl concentration had no significant effect on lipid oxidation. However, when the NaCl concentration was further increased to exceed 300 mM, the CD value of filled hydrogels exceeded that of filled hydrogels without NaCl (*p* < 0.05), suggesting that relatively higher NaCl concentrations increased the rate of lipid oxidation by increasing the catalytic activity of iron [44].

TBARS values are indicators of the production of secondary oxidation products during the termination phase of lipid oxidation. As shown in Figure 8B, although the TBARS values of freshly prepared filled hydrogels increased when the NaCl concentration increased from 0 to 500 mM, there was no significant difference for each filled hydrogel (*p* < 0.05). Moreover, after 10 days, TBARS values of filled hydrogels obviously decreased and then increased with increasing NaCl concentrations (*p* < 0.05). Similar results were observed by Zhu et al. [45], who reported that emulsions had lower TBARS values at low NaCl concentrations and higher TBARS values at higher NaCl concentrations compared with those without NaCl. Therefore, the above results indicated that adding NaCl into the filled hydrogel directly influences the lipid oxidative stability to some extent.

### 3.9. Protein Oxidation

The initial level of natural tryptophan fluorescence displayed by filled hydrogels depended on the NaCl concentration. As seen in Figure 9A, with an increase in storage time from 0 to 10 days, the tryptophan fluorescence intensity gradually decreased (*p* < 0.05). The reduction in tryptophan fluorescence is indicative of the oxidative degradation of tryptophan, which is related to a metal such as copper [46]. Moreover, the fluorescence intensity of filled hydrogels decreased and then tended to decrease with an increase in NaCl concentration with the highest fluorescence intensity recorded at 200 mM NaCl for the same storage time. Adding NaCl causes the structure of proteins to loosen, which may be conducive to the exposure of tryptophan and increase the tryptophan fluorescence intensity [47]. However, a relatively higher NaCl concentration inhibited the electrostatic repulsion between the adsorbed film and the biopolymers, which may promote extensive protein aggregation and increase steric hindrance, thereby quenching the fluorescence of tryptophan.

FP was a secondary protein oxidation that could usually account for the degradation of protein oxidation. As seen in Figure 9B, the fluorescence emitted by FP increased over time, with the highest fluorescence value being observed at 10 days. Corresponding to the quenching of tryptophan fluorescence, increasing fluorescence for FP may be due to interactions between oxidized proteins and secondary lipid oxidation products [48]. Moreover, a significant difference was observed between filled hydrogels with and without NaCl, wherein FP first increased and then decreased with increasing NaCl concentration at each individual storage time. This was because NaCl addition promoted a reaction between free amino groups from the proteins and secondary oxidation products, leading to the formation of FP. With a further increase in NaCl concentration, the FP may aggregate at certain conditions and decrease the fluorescence intensity. Therefore, this result indicated that the addition of NaCl had an obvious concentration-dependent influence on protein oxidation.

Based on above results of lipids oxidation and protein oxidation, the effects of NaCl on lipids oxidation and protein oxidation of filled hydrogels are shown in Figure 10. The addition of NaCl could increase the catalytic activity of iron and promote the dissociation of the H atom of the adjacent methylene carbon atom of the lipid double bond, thus leading to the formation of primary lipid oxidation products and secondary lipid oxidation products. Moreover, NaCl could reduce the negative charge of the filled hydrogel and weaken the electrostatic adsorption force between the positively charged metal ions, resulting in a weakening of the antioxidant capacity of protein and promoting the oxidation reaction of the droplets [45]. In addition, based on the microstructure analysis, the protein aggregation into continuous phase decreased the antioxidant capacity, thus the oxidation of unsaturated fatty acids under certain conditions is promoted. For the protein oxidation, NaCl addition led to the extensive protein aggregation and increased steric hindrance, decreasing the ABTS^+^ free radical scavenging ability, metal ion chelating ability and reducing ability, thus promoting the transformation of tryptophan residues that are located in the internal surface of the droplets to form radicals and the formation of protein oxidation products. In addition, the addition of NaCl promoted the formation of primary lipid oxidation products and secondary lipid oxidation products, which may have contributed to the reaction with tryptophan peroxyl radicals and accelerated the oxidation of the protein [48].

### 3.10. Correlation Analyses

Correlation analysis is an effective technology to evaluate the relationship between physical and oxidative stabilities upon storage up to 10 days. As shown in Figure 11, the particles size (*D_4,3_*) and zeta-potential of fresh filled hydrogels had positive correlations with TBARS values (R = 0.82 and 0.85). The results were similar to those of our previous study [19], which reported that the droplet size was negatively correlated with TBARS value for filled hydrogels. Ling et al. [17] also noted that the lipid oxidation products may affect the charge behavior of the emulsions. Moreover, no significant correlation was found between the CD value and tryptophan fluorescence spectroscopy (R = −0.03), whereas a significant correlation was found between CD value and FP (R = 0.5). Furthermore, TBARS value was negatively correlated with tryptophan fluorescence spectroscopy and FP (R = −0.71 and −0.77). Kagan [49] indicated that the free amino groups could react with secondary oxidation products of lipids and result in promoting the formation of protein oxidation products. After 10 days, the parameters of filled hydrogels had lower values than those for filled hydrogels in storage on day 0, indicating that prolonging storage time could decrease the physical and oxidative stabilities of filled hydrogels. Notably, CD and TBARS values were more negatively correlated to tryptophan fluorescence spectroscopy and FP at 10 days compared with filled hydrogels on day 0. This indicated that a more notable correlation in protein and lipid oxidation was observed with extended storage time.

## 4. Conclusions

The present work studied the effects of NaCl level on the physical and oxidative stabilities of filled hydrogels. The addition of NaCl decreased the physical stability of filled hydrogels, which was mainly ascribed to the electrostatic shielding between droplets or the weak electrostatic attraction between protein and polysaccharide molecules around the droplet surfaces, eventually leading to coalescence or creaming of droplets with large average particle sizes. Moreover, there was no significant difference for lipid oxidation at lower NaCl concentrations, whereas higher NaCl concentrations extensively enhanced protein aggregation and loosened the structure of proteins, which may be conducive to lipid as well as protein oxidation. Based on these results, we conclude that the physical and oxidative stabilities of filled hydrogels strongly depended on the NaCl concentration. Future research should study the effects of other environmental conditions, such as pH, temperature, and shear rate, on the stability of filled hydrogels.

## Figures and Tables

**Figure 1 foods-10-01027-f001:**
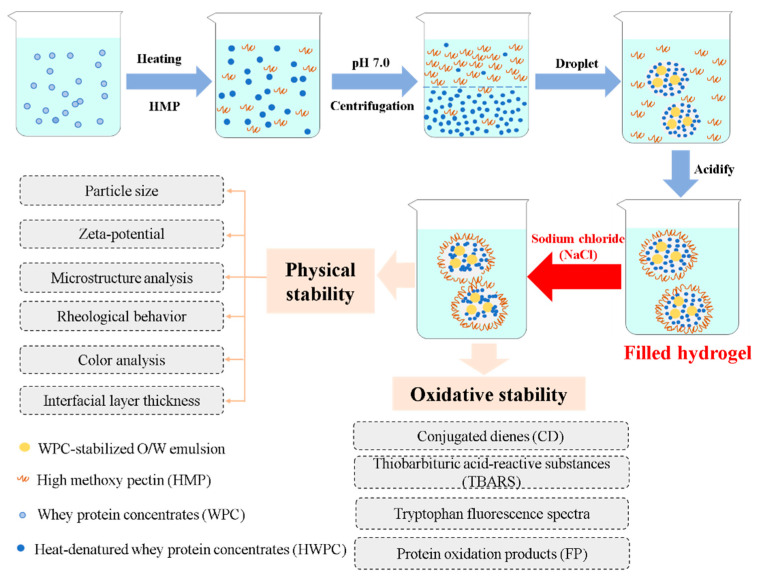
Schematic diagram of fabrication of filled hydrogels at different sodium chloride (NaCl) concentrations.

**Figure 2 foods-10-01027-f002:**
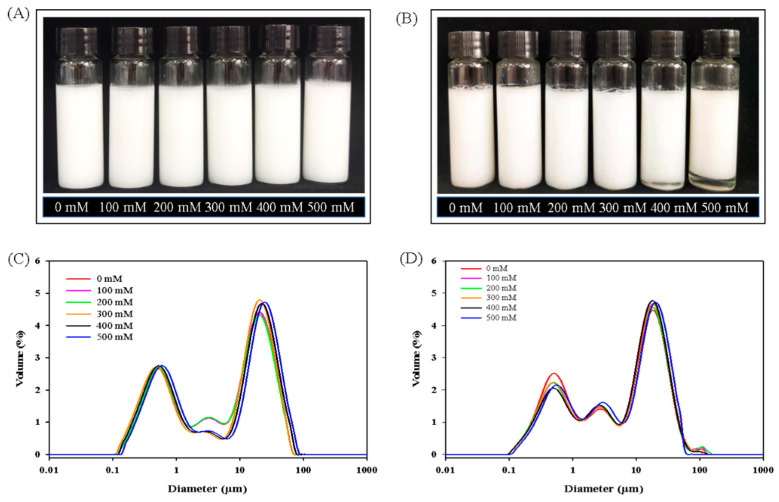
Effects of NaCl concentration on the appearance (**A**,**B**) and particle size distribution (**C**,**D**) of filled hydrogels at stored for 0 days (**A**,**C**) for 10 days (**B**,**D**).

**Figure 3 foods-10-01027-f003:**
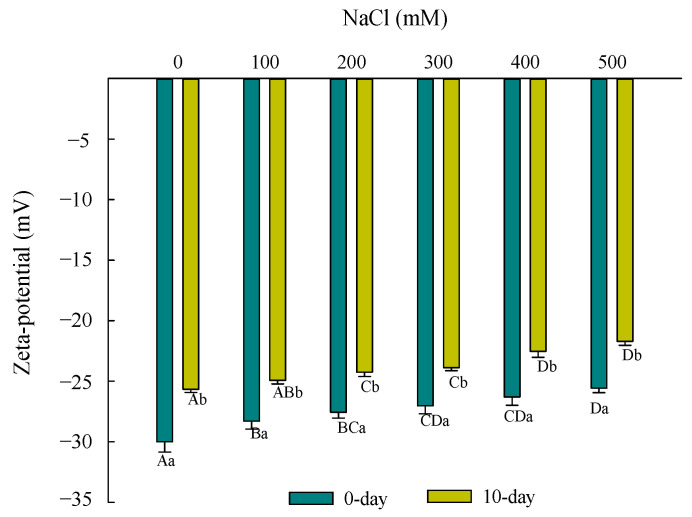
Effects of NaCl concentration on the zeta-potential of filled hydrogels during storage. Different letters from A to D indicate significant differences for different NaCl concentration at same days (*p* < 0.05); Different letters from a to b indicate significant differences for different days at same NaCl concentration (*p* < 0.05).

**Figure 4 foods-10-01027-f004:**
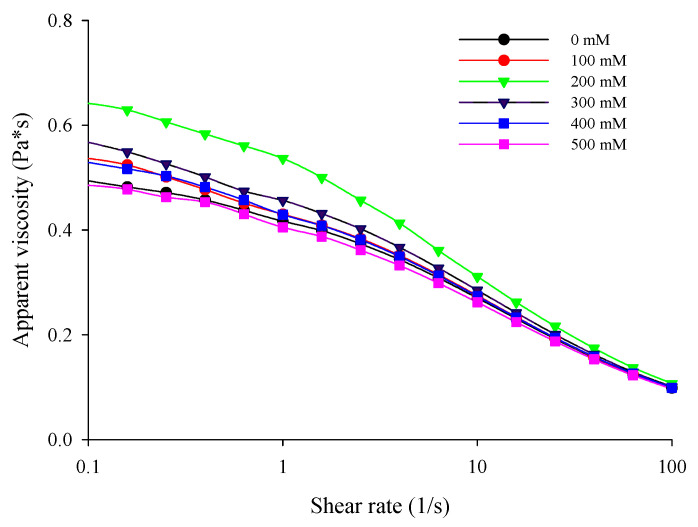
Effects of NaCl concentration on the apparent viscosity of filled hydrogels.

**Figure 5 foods-10-01027-f005:**
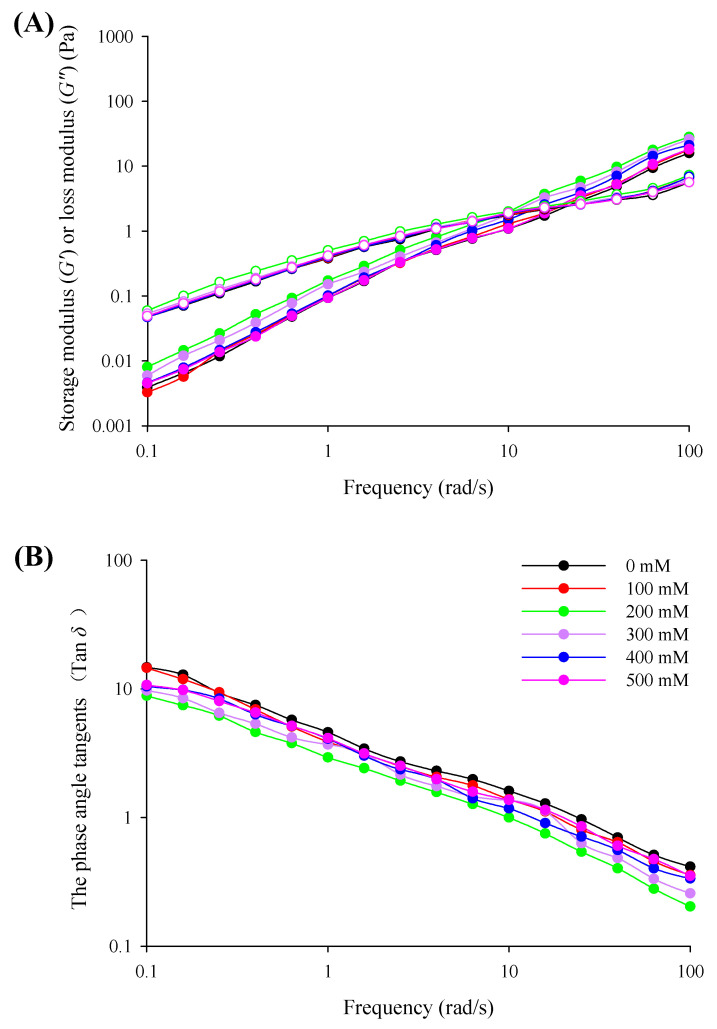
Frequency dependency of storage modulus (*G*′) or loss modulus (*G*″) (**A**), and the phase angle tangents (Tan *δ*) (**B**) for filled hydrogels at different NaCl concentrations.

**Figure 6 foods-10-01027-f006:**
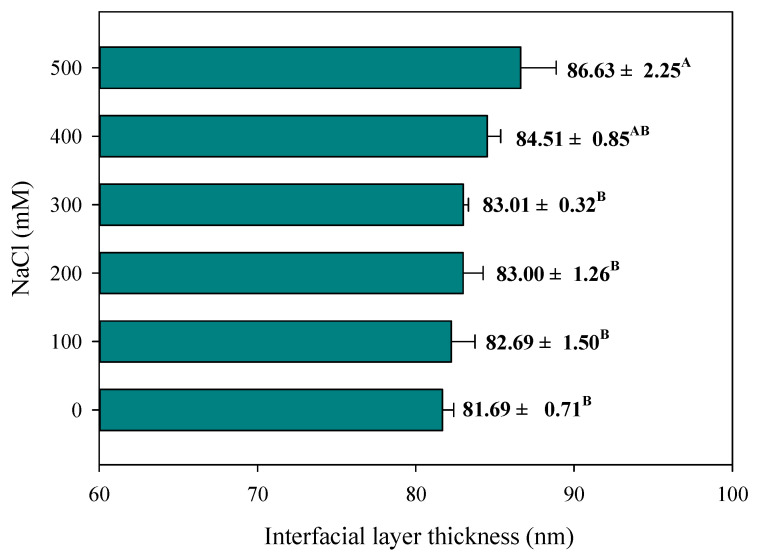
Effects of NaCl concentration on the interfacial layer thickness of filled hydrogels. Different letters from A to B indicate significant differences at different NaCl concentration (*p* < 0.05).

**Figure 7 foods-10-01027-f007:**
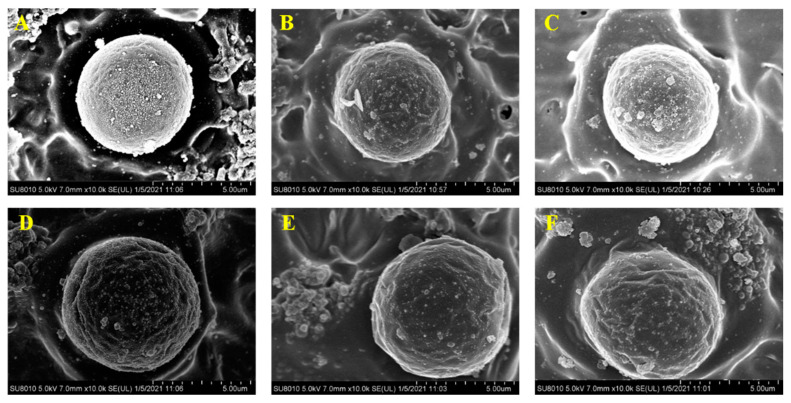
Cryo-SEM images of filled hydrogels treated with different NaCl concentrations. (**A**–**F**) represented the NaCl concentration of 0, 100, 200, 300, 400, 500 mM.

**Figure 8 foods-10-01027-f008:**
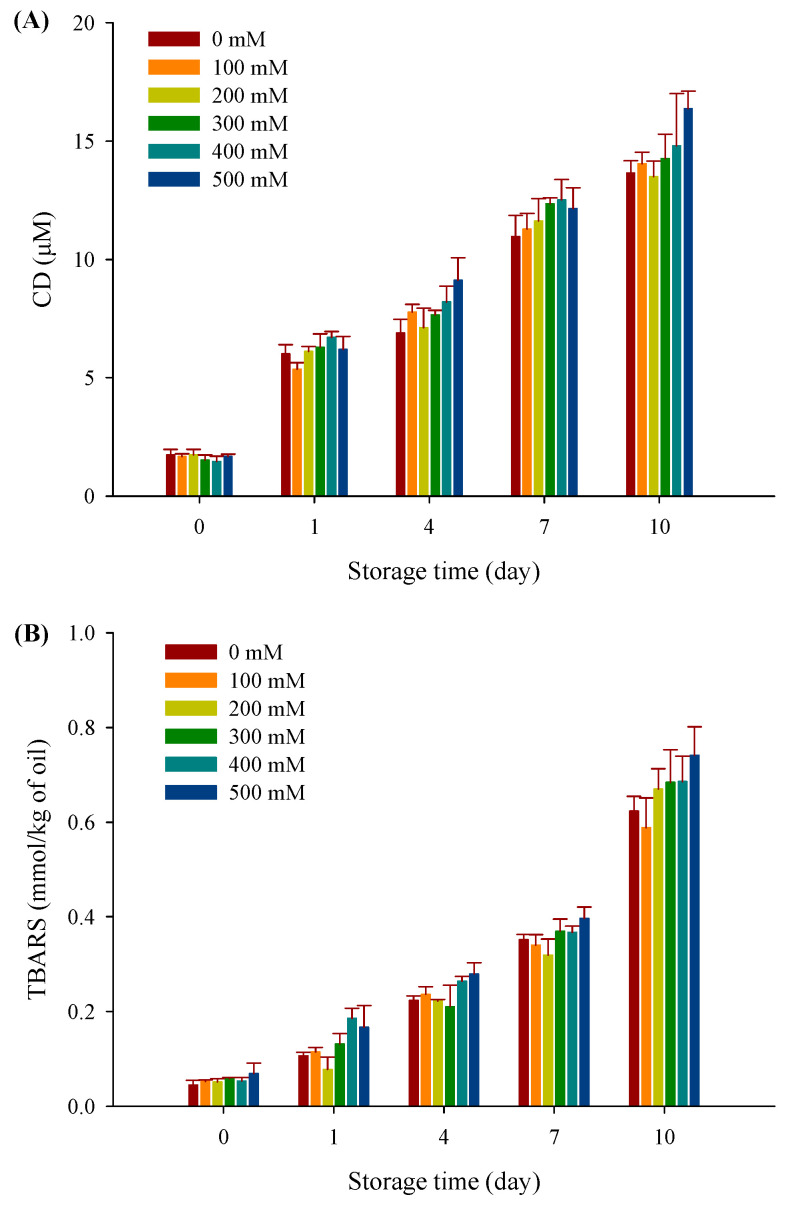
CD values (**A**) and TBARS values (**B**) for filled hydrogels at different NaCl concentrations stored at 37 °C for 10 days.

**Figure 9 foods-10-01027-f009:**
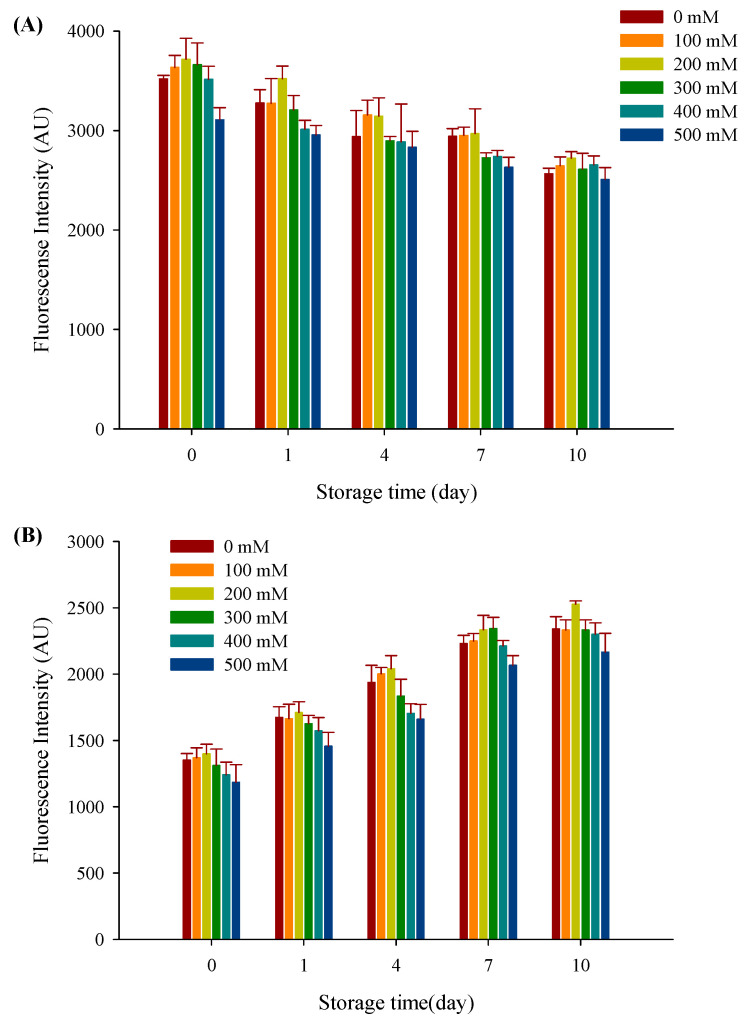
The fluorescence spectra of tryptophan (**A**) and protein oxidation products (**B**) of filled hydrogels treated with different NaCl concentrations at 37 °C during storage.

**Figure 10 foods-10-01027-f010:**
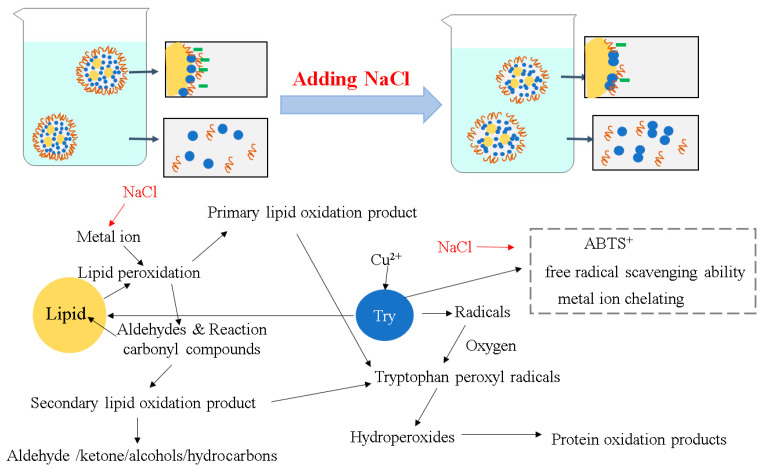
The pictorial mechanism of NaCl on the lipid and protein oxidation of filled hydrogels.

**Figure 11 foods-10-01027-f011:**
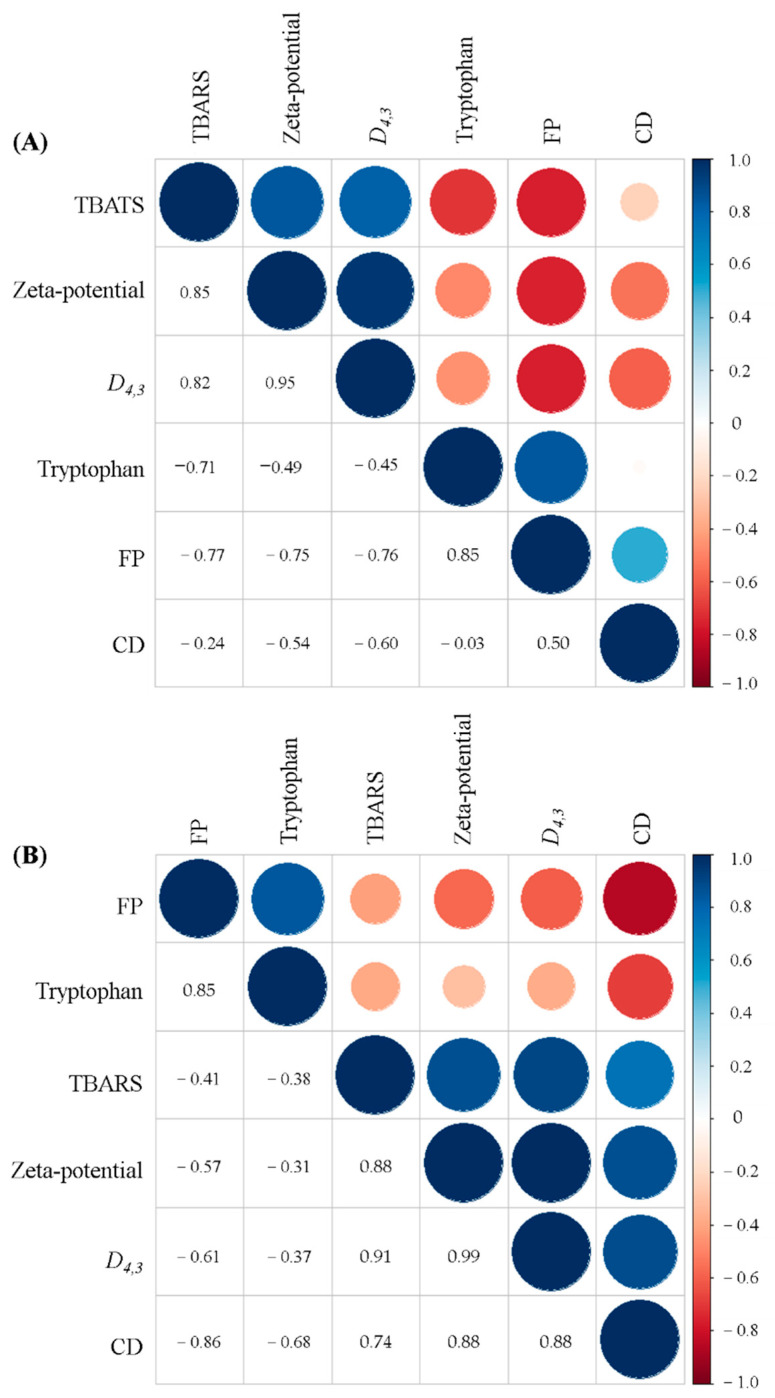
Correlation analysis of filled hydrogels for the physical and oxidative stability at different NaCl concentrations stored for 0 days (**A**) and 10 days (**B**).

**Table 1 foods-10-01027-t001:** Effects of sodium chloride (NaCl) concentration on droplet size of filled hydrogels during storage.

Sodium Chloride (NaCl, mM)	The Volume Averaged Diameter (*D_4,3_*, μm)		The Surface-Averaged Diameter (*D_3,2_,* μm)	
	Day 0	Day 10	Day 0	Day 10
0	23.60 ± 0.42 ^D,b^	26.60 ± 0.28 ^E,a^	4.82 ± 0.06 ^D,b^	8.19 ± 0.06 ^F,a^
100	23.99 ± 0.64 ^D,b^	28.24 ± 0.95 ^D,a^	4.89 ± 0.07 ^D,b^	8.87 ± 0.11 ^E,a^
200	26.61 ± 0.20 ^C,b^	29.69 ± 0.71 ^C,D,a^	5.28 ± 0.10 ^C,b^	9.17 ± 0.14 ^D,a^
300	28.91 ± 0.86 ^B,b^	31.16 ± 0.72 ^C,a^	5.956 ± 0.06 ^B,b^	9.84 ± 0.06 ^C,a^
400	29.70 ± 0.41 ^B,b^	34.67 ± 0.42 ^B,a^	6.17 ± 0.09 ^A,b^	10.23 ± 0.11 ^B,a^
500	31.18 ± 0.34 ^A,b^	37.51 ± 0.54 ^A,a^	6.38 ± 0.11 ^A,b^	11.37 ± 0.10 ^A,a^

Values are given as means ± standard deviations (SD) from triplicate determinations; ^A–F^ in each column represent statistically significant differences (*p* < 0.05); ^a–b^ in each line represent statistically significant differences (*p* < 0.05).

**Table 2 foods-10-01027-t002:** Effects of NaCl concentration on the steady shear parameter of filled hydrogels.

NaCl (mM)	Zero-Shear Viscosity (*η*_0_)	Time Constant (*λ*)	Dimensionless Exponent (*m*)	Correlation Coefficient (R^2^)
0	0.4939 ± 0.0011 ^D^	0.0758 ± 0.0012 ^D^	0.7427 ± 0.0021 ^B^	0.9888
100	0.5367 ± 0.0037 ^C^	0.0957 ± 0.0002 ^B^	0.7069 ± 0.0018 ^D^	0.9713
200	0.6418 ± 0.0012^A^	0.1040 ± 0.0017 ^A^	0.7056 ± 0.0022 ^D^	0.9993
300	0.5673 ± 0.0151 ^B^	0.1010 ± 0.0015 ^A^	0.6898 ± 0.0041 ^E^	0.9834
400	0.5290 ± 0.0110 ^C^	0.0897 ± 0.0021 ^C^	0.7334 ± 0.0029 ^C^	0.9786
500	0.4653 ± 0.0102 ^E^	0.0785 ± 0.0009 ^D^	0.7673 ± 0.0011 ^A^	0.9737

Values are given as means ± SD from triplicate determinations; ^A–E^ in each column represent statistically significant differences (*p* < 0.05).

**Table 3 foods-10-01027-t003:** Parameters of power law model in frequency sweep tests for filled hydrogels with different NaCl concentrations.

NaCl (mM)	*log G*′ = *log a*′ + *b*′ *log f*			*log G*″ = *log a*″ + *b*″ *log f*		
	Model Parameters (*a*′)	Dimensionless Frequency Indices (*b*′)	R^2^	Model Parameters (*a*″)	Dimensionless Frequency Indices (*b*″)	R^2^
0	0.0746 ± 0.0039 ^D^	1.2236 ± 0.0026 ^A^	0.9937	0.3170 ± 0.0211 ^B^	0.6827 ± 0.0022 ^A^	0.9764
100	0.0769 ± 0.0011 ^D^	1.2103 ± 0.0031 ^B^	0.9925	0.3290 ± 0.0165 ^B^	0.6798 ± 0.0042 ^A^	0.9792
200	0.1448 ± 0.0058 ^A^	1.1673 ± 0.0023 ^D^	0.9979	0.4126 ± 0.0219 ^A^	0.6438 ± 0.0026 ^C^	0.9777
300	0.1176 ± 0.0037 ^B^	1.1878 ± 0.0018 ^C^	0.9975	0.3497 ± 0.0179 ^B^	0.6609 ± 0.0031 ^B^	0.9789
400	0.0890 ± 0.0033 ^C^	1.1897 ± 0.0009 ^C^	0.9971	0.3266 ± 0.0204 ^B^	0.6695 ± 0.0024 ^B^	0.9798
500	0.0795 ± 0.0038 ^C,D^	1.1861 ± 0.0012 ^C^	0.9961	0.3364 ± 0.0192 ^B^	0.6640 ± 0.0039 ^B^	0.9748

Values are given as means ± SD from triplicate determinations; ^A–D^ in each column represent statistically significant differences (*p* < 0.05).

**Table 4 foods-10-01027-t004:** Effects of NaCl concentration on the color of filled hydrogels.

NaCl (mM)	*L** Value	*a** Value	*b** Value	Whiteness
0	59.86 ± 0.48 ^A^	−2.00 ± 0.05 ^A^	−2.75 ± 0.29 ^A^	59.82 ± 0.12 ^A^
100	58.86 ± 0.70 ^A,B^	−2.27 ± 0.60 ^A^	−2.54 ± 0.29 ^A^	58.72 ± 0.13 ^B^
200	57.53 ± 0.61 ^B,C^	−2.18 ± 0.40 ^A^	−2.75 ± 0.20 ^A^	57.39 ± 0.21 ^C^
300	57.31 ± 0.50 ^C,D^	−2.32 ± 0.17 ^A^	−1.89 ± 0.10 ^B^	57.22 ± 0.14 ^C^
400	56.56 ± 0.27 ^C,D^	−1.98 ± 0.12 ^A^	−2.56 ± 0.08 ^A^	56.44 ± 0.27 ^D^
500	56.08 ± 0.27 ^D^	−2.04 ± 0.05 ^A^	−2.89 ± 0.06 ^A^	55.94 ± 0.32 ^D^

Values are means ± SD from triplicate determinations; ^A–D^ in each column represent statistically significant differences (*p* < 0.05).

## Data Availability

The data presented in this study are available in the article.

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
