# Peer review of "Effects of Sodium Chloride on the Physical and Oxidative Stability of Filled Hydrogel Particles Fabricated with Phase Separation Behavior"

_foods, 2021, doi:10.3390/foods10051027_

Round 1

Reviewer 1 Report

The paper “Effects of Sodium Chloride on the Physical and Oxidative Stability of Filled Hydrogel Particles Fabricated with Phase Separation Behaviour" is focused on a very interesting and useful topic. In addition, they extend the knowledge about novel delivery systems for specific applications in food products.  However, I think that authors should extend the introduction and pay more attention to the rheological results.

I have some comments:

Line 21. Authors have used L* in the abstract without definition. Abstract should be understood alone.

Line 113. “using a controlled stress/strain rheometer”. To the best of my knowledge, a rheometer can be controlled stress or controlles strain but not both.

Line 114. Authors do not use a serrated or sand-blasted geometry for measuring particles. Slip effects could take part in the rheological tests.

Line 125. It is elastic modulus and viscous modulus. In singular.

Line 241. In my opinion, authors should try other models such as Cross model since flow curves have a tendency to reach a zero-shear viscosity at low shear rates.

Line 243. Thixotropic is not related to flow index. Please, remove this. It is a very big mistake. If authors want to know about the thixotropy of the samples, they have to carry out hysteresis loops, for example.

Line 250. Authors do not explain why viscosity increases from 0 to 200 mM.

Figure 2A and 2B. G’ and G’’ are usually plotted in the same graph in order to compare their values. This are called mechanical spectra.

Author Response

Q1: Line 21. Authors have used L* in the abstract without definition. Abstract should be understood alone.

A1: This is a good suggestion. According to the reviewer’s opinion, we have revised the abstract of our paper as follows:

“Our results showed that with an increase in NaCl concentration, the particle sizes, zeta-potentials, and interfacial layer thickness of filled hydrogels significantly increased and the lightness and whiteness gradually decreased (P < 0.05).”

We are indebted to the reviewer for this constructive suggestion to improve the quality of the manuscript. Thanks.

Q2: Line 113. “using a controlled stress/strain rheometer”. To the best of my knowledge, a rheometer can be controlled stress or controlles strain but not both.

A2: This is a good question. First of all, we were apologized for the incorrect statement in our initial manuscript. In fact, in our experience, we chose the controlled stress rheometer to evaluate the rheological properties. In our revised manuscript, we have rewritten this statement as follows:

In the section of “Materials and Methods

2.6. Rheological behavior

Two types of oscillatory measurements, steady shear and frequency sweep tests, were conducted using a controlled stress rheometer (DHR-1, TA Instruments, Delaware, USA) equipped with serrated plate-plate geometry (40 mm diameters) and a gap between parallel plates of 1 mm.

We are indebted to the reviewer for this constructive suggestion to improve the quality of the manuscript. Thanks.

Q3: Line 114. Authors do not use a serrated or sand-blasted geometry for measuring particles. Slip effects could take part in the rheological tests.

A3: This is a good question. First of all, we were apologized for the unclear statement in our initial manuscript. In fact, in our experience, we use the serrated geometry for measuring filled hydrogels particles. In our revised manuscript, we have rewritten this statement as follows:

In the section of “Materials and Methods

2.6. Rheological behavior

Two types of oscillatory measurements, steady shear and frequency sweep tests, were conducted using a controlled stress rheometer (DHR-1, TA Instruments, Delaware, USA) equipped with serrated plate-plate geometry (40 mm diameters) and a gap between parallel plates o 1 mm.

We are indebted to the reviewer for this constructive suggestion to improve the quality of the manuscript. Thanks.

Q4: Line 125. It is elastic modulus and viscous modulus. In singular.

A4: We have revised it as suggested. Thanks.

Q5: Line 241. In my opinion, authors should try other models such as Cross model since flow curves have a tendency to reach a zero-shear viscosity at low shear rates.

A5: This is a good suggestion. According to the reviewer’s opinion, we have analyzed the data of steady shear test by using the Cross model. In our revised manuscript, we have added some statements as follows:

In the section of “Materials and Methods

2.6.1. Steady shear tests

Apparent viscosities curves were constructed through continuous steady shear tests varying the shear rate from 0.1 to 100 s-1. The obtained curves were fitted to the Cross model that was used to evaluate the dependence relationship between shear rate and apparent viscosity as follows:

=                              (1)

Where η represents apparent viscosity (Pa.s); η0 represents zero-shear viscosity (Pa.s) at low shear rate; η represents infinite-shear viscosity (Pa.s) at the high shear rate; γ represents the shear rate (s-1); λ represents the time constant (s); m represents the dimensionless exponent.

In the section of “Results and discussion

3.3. Apparent viscosity

Evaluating the flow behaviour of filled hydrogel particles was important because it could well reflect their practical application in the manufacture of food products. As shown in Fig. 4, there was an obviously decrease in apparent viscosity as the shear rate was increased from 0.1 to 100 s-1 for all the filled hydrogels, suggesting a typical shear thinning flow behaviour. As shown in Table 2, the correlation coefficient (R2) for each filled hydrogel was above 0.97, indicating that the flow curve was accurately described by the Cross model. Moreover, the value of dimensionless exponents (m) tends to 1, which also confirmed a non-Newtonian shear-thinning behaviour. The main reason for this behaviour was the destruction of molecular aggregation and arrangement caused by an increasing shear rate; the destruction rate was higher than the reintegration rate, thereby leading to transformational behaviour from high viscosity to low viscosity [27, 28, 29].

In Cross model, zero-shear viscosity (η0) increased when the NaCl concentration increased from 0 to 200 mM (P < 0.05), indicating that the addition of NaCl increase the resistance to flow due to the interaction. Moreover, with an increase in the concentration of NaCl from 0 to 200 mM, the apparent viscosity of filled hydrogels increased. Similar results were observed in the study by Shao et al. [30], who reported that the addition of NaCl could increase the viscosity of hydrocolloids and stabilized the emulsion; thus, the viscosity of the emulsion varied with the shear rate. Griffin et al. [31] found that after the adsorption of xanthan gum onto the surfaces of whey protein-coated droplets, the addition of NaCl promoted the bridge between droplets owing to the electrostatic screening effect that contributed to the association between protein-coated droplets and further increasing the viscosity. In addition, Liu et al. reported that the addition of NaCl increased the viscosity of emulsion, which confirmed that non-adsorbed biopolymer (such as protein or polysaccharide) in emulsion after the addition of NaCl may be enhance the viscosity in aqueous phase, resulting in astricting the movement of droplets with a higher viscosity [32]. However, the apparent viscosity of filled hydrogels slowly decreased when the NaCl concentration further increased from 300 to 500 mM. Meanwhile, η0 decreased with the increase of NaCl concentration (P < 0.05). Similar phenomenon was observed in a previous study by Sriprablom et al. [33], who stated demonstrated the effect NaCl concentration on the rheological properties of an emulsion stabilized by whey protein and xanthan gum. The flow behaviour of proteins and polysaccharides at different NaCl concentrations may determine the apparent viscosity of filled hydrogels. Cai et al. [27] reported that the structure of xanthan gum transformed from ordered (helix) to disordered (coil) when NaCl was added, thereby changing the flow behaviour with low apparent viscosity. Moreover, Hao et al. [34] found that increasing NaCl concentration caused the charge shielding effect and made the polysaccharide molecules shrink, thus expressing a lower viscosity. Notably, the time constant (λ) increased and then decreased with increasing NaCl concentration in Table 2, which also verified aforementioned apparent viscosity phenomenon.

We are indebted to the reviewer for this constructive suggestion to improve the quality of the manuscript. Thanks.

Q6: Line 243. Thixotropic is not related to flow index. Please, remove this. It is a very big mistake. If authors want to know about the thixotropy of the samples, they have to carry out hysteresis loops, for example.

A6: This is a good question. First of all, we were apologized for the unclear statement in our initial manuscript. In fact, we just want to illustrate the shear-thinning behaviour via shear rate of sample rather than the thixotropic. According to the reviewer’s opinion, we have deleted the thixotropic statements as follows:

“As shown in Table 2, the correlation coefficient (R2) for each filled hydrogel was above 0.97, indicating that the flow curve was accurately described by the cross model. Moreover, the value of dimensionless exponents (m) tends to 1, which also confirmed a non-Newtonian shear-thinning behaviour.”

We are indebted to the reviewer for this constructive suggestion to improve the quality of the manuscript. Thanks.

Q7: Line 250. Authors do not explain why viscosity increases from 0 to 200 mM.

A7: This is a good question. The reasons why the viscosity of filled hydrogels increased with increasing NaCl concentration from 0 to 200 mM were as follows: First of all, in the filled hydrogels, after the adsorption of xanthan gum onto the surfaces of whey protein-coated droplets, the addition of NaCl promoted the bridge between droplets owing to the electrostatic screening effect that contributed to the association between protein-coated droplets and further increasing the viscosity. Moreover, non-adsorbed biopolymer (such as protein or polysaccharide) in emulsion after the addition of NaCl may be enhance the viscosity in aqueous phase, resulting in astricting the movement of droplets with a higher viscosity in emulsion. In addition, according to the study of Shao et al. (Food Hydrocolloid. 2017, 69, 202-209), Griffin et al. (Journal of Food Engineering, 2020, 277, 10989), Liu et al. (Food Hydrocolloid. 2020, 100, 105431), the addition of NaCl could increase the viscosity of the emulsion. According to the reviewer’s opinion, we have added some statement about the reason of viscosity increases from 0 to 200 mM of this work as follows:

“Similar results were observed in the study by Shao et al. [30], who reported that the addition of NaCl could increase the viscosity of hydrocolloids and stabilized the emulsion; thus, the viscosity of the emulsion varied with the shear rate. Griffin et al. [31] found that after the adsorption of xanthan gum onto the surfaces of whey protein-coated droplets, the addition of NaCl promoted the bridge between droplets owing to the electrostatic screening effect that contributed to the association between protein-coated droplets and further increasing the viscosity. In addition, Liu et al. reported that the addition of NaCl increased the viscosity of emulsion, which confirmed that non-adsorbed biopolymer (such as protein or polysaccharide) in emulsion after the addition of NaCl may be enhance the viscosity in aqueous phase, resulting in astricting the movement of droplets with a higher viscosity [32].”

The five newly cited references are as follows:

32     Liu, Y.; Hu, X.B.; Ye, Y.F.; Wang, M.M.; Wang, J.H. Emulsifying properties of wheat germ: Influence of pH and NaCl. Food Hydrocolloid. 2020, 100, 105431. https://doi.org/10.1016/j.foodhyd.2019.105431

We are indebted to the reviewer for this constructive suggestion to improve the quality of the manuscript. Thanks.

Q8: Figure 2A and 2B. G’ and G’’ are usually plotted in the same graph in order to compare their values. This are called mechanical spectra.

A8: This is a good suggestion. According to the reviewer’s opinion, we have repainted the mechanical spectra of G’ and G’’. We are indebted to the reviewer for this constructive suggestion to improve the quality of the manuscript. Thanks.

Reviewer 2 Report

Manuscript entitled “Effects of Sodium Chloride on the Physical and Oxidative Stability of Filled Hydrogel Particles Fabricated with Phase Separation Behaviour” described the effects of NaCl on the physical and oxidative stability of filled hydrogel. This manuscript could be interesting for the readers. However, the paper needs a significant revision. I have listed a few comments that need to be addressed:

  1. What is the novelty and importance of this work?
  2. Authors are advised to add a schematic illustration to show the overall work.
  3. In introduction add more up-to-date information. Very limited information is provided regarding whey protein concentrate and high methoxy pectin.
  4. For more insight into this work, I would advise to add the color parameter a, b, and whiteness index in section 2.8.
  5. Why whey protein concentrates and high methoxy pectin was chosen? Justify.
  6. Add the size and zeta potential graph.
  7. How about the effect of morphology of the hydrogel? Add SEM image.
  8. Add apparent image of all the hydrogel.
  9. Authors are advised to add the pictorial mechanistic aspect of NaCl on the lipid and protein oxidation and discuss it in the text.
  10. The discussion could be more solid. Revise it carefully.
  11. Also, carefully revise the manuscript for linguistic and typos errors.

Author Response

Q1: What is the novelty and importance of this work?

A1: This is a good question. First of all, the prepared filled hydrogels have the potential to provide benefits such as improved stability, effective controlled release of bioactive compounds, and improved overall flavor intensity for the reduced fat emulsions system when compared with other conventional emulsions. Moreover, emulsion-based systems were highly sensitive to droplets aggregation when the various food processing conditions such as changes in ionic strength, pH, temperature, and mechanical mixing speed during manufacture of commercial food or medicinal products were altered, which would limit the utilization of filled hydrogels in many food products. Thereby, the investigation of the influence of NaCl concentration on the physical and oxidative stabilities of filled hydrogel particles was necessary and important, which will contribute to the design and manufacture of novel delivery systems for specific applications at various food processing conditions especially for salt condition in a range of functional food products. In our manuscript, we describe the novelty and importance of this work.

We are indebted to the reviewer for this constructive suggestion to improve the quality of the manuscript. Thanks.

Q2: Authors are advised to add a schematic illustration to show the overall work.

A2: This is a good suggestion. According to the reviewer’s opinion, we have added a schematic illustration of this work as follows:

In the section of “2.4 Filled hydrogel preparation

“The preparation procedures of filled hydrogels at difference NaCl concentrations are shown in Fig. 1. The filled hydrogel particles were prepared according to the procedure described by Matalanis et al. [17] with a few modifications.”

We are indebted to the reviewer for this constructive suggestion to improve the quality of the manuscript. Thanks.

Q3: In introduction add more up-to-date information. Very limited information is provided regarding whey protein concentrate and high methoxy pectin.

A3: This is a good suggestion. According to the reviewer’s opinion, we have added some information about the whey protein concentrate and high methoxy pectin in “Introduction” as follows:

In the section of “Introduction

“Whey protein concentrate obtained by removing non-protein components from pasteurized whey. After heating over denaturation temperatures, whey protein will ex-pose more hydrophobic groups and sulfhydryl groups buried in the molecule due to the partially unfold the globular protein. Moreover, as a natural anionic polysaccharide, HMP has been widely used in the food industry because of good gelling and thickening properties. Therefore, the change of protein structure and electrostatic interaction between HWPC and HMP at neutral pH would attribute to the occur of phase separation behavior and the formation of filled hydrogels.”

We are indebted to the reviewer for this constructive suggestion to improve the quality of the manuscript. Thanks.

Q4: For more insight into this work, I would advise to add the color parameter a, b, and whiteness index in section 2.8.

A4: This is a good suggestion. According to the reviewer’s opinion, we have added the color parameter a, b, and whiteness index and discuss it in our revised manuscript as follows:

In the section of “Materials and Methods

2.8. Color measurement

Before testing, the color meter was standardized using white and black calibration plates. The color (L*, a*, b*) of each sample was measured based on the procedure described in our previous study [19]. Whiteness was calculated using the equation (4) as follows:

Whiteness = 100 -          (4)

In the section of “Results and discussion

In addition, the whiteness of filled hydrogels increased with increasing NaCl concentration, which was in accordance with the results of L* value. It is noted that there were no statistically significances for a* value and b* value, indicating that the addition NaCl had almost no effect on the redness and yellowness of filled hydrogels.

We are indebted to the reviewer for this constructive suggestion to improve the quality of the manuscript. Thanks.

Q5: Why whey protein concentrates and high methoxy pectin was chosen? Justify.

A5: This is a good question. The reasons why we use whey protein concentrates (WPC) and high methoxy pectin (HMP) to prepare filled hydrogels were as follows: Firstly, WPC obtained by removing non-protein components from pasteurized whey. Moreover, as a natural anionic polysaccharide, HMP has been widely used in the food industry because of good gelling and thickening properties. The electrostatic interaction between WPC and HMP at neutral pH may be attribute to the occur of phase separation behavior and the formation of filled hydrogels. Moreover, in our pre-experiment, we found that the mixtures of WPC and HMP was not occur phase separation behavior. However, the phase separation behavior occurred when mixing heat-denatured whey protein concentrate (HWPC) and HMP solutions at different mass ratios. Whey protein molecules were prone to forming long and thin fibrous structures at neutral pH when heated over their thermal denaturation temperatures. Therefore, the change of protein structure and electrostatic interaction between HWPC and HMP at neutral pH would attribute to the occur of phase separation behavior and the formation of filled hydrogels. In addition, some related references reported the formation of multiple emulsions based on thermodynamic incompatibility of heat-denatured whey protein and pectin solutions, such as Kim et al. (Food Hydrocolloids, 2006, 20, 586-595). Thus, based on some results of previous studies and our pre-experiment, we finally select the HWPC and HMP.

We are indebted to the reviewer for this constructive suggestion to improve the quality of the manuscript. Thanks.

Q6: Add the size and zeta potential graph.

A6: This is a good question. According to the reviewer’s opinion, we have added the data of particle size distribution of filled hydrogels and the related statement in section of results and discussion as follows:

In the section of “Results and discussion

“An aim of this experiment was to examine the influence of NaCl concentration on the physical stability of filled hydrogels. All freshly prepared filled hydrogels showed a uniform and white appearance in Fig 2A, which was supported the whiteness results in colour analysis. Moreover, the particle size distributions of freshly prepared filled hydrogels at difference NaCl concentration are shown in Fig. 2C. The large particle peak in the particle size distribution of filled hydrogels shifted toward the right with the increase of NaCl level increased from 0 to 500 mM, suggesting an increase of particle size. Meanwhile, Table 1 shows that as the NaCl level increased from 0 to 500 mM, the volume averaged diameter (D4,3) and the surface-averaged diameter (D3,2) of the freshly prepared filled hydrogel increased (P < 0.05). For instance, when compared with filled hydrogel without NaCl, the D4,3 of filled hydrogel increased by 1.63, 12.75, 22.50, 25.85, and 32.12% for NaCl concentrations of 100, 200, 300, 400, and 500 mM, respectively. This phenomenon was mainly attributed to electrostatic screening, ion binding, interfacial rearrangements, or polysaccharide desorption [21, 22]. Harnsilawat et al. [23] reported the gradual screening of electrostatic repulsion among droplets gradually increased with increasing NaCl concentration, eventually leading to the coalescence or creaming of droplets. Moreover, the presence of NaCl weakened the electrostatic attraction between the protein and polysaccharide molecules around the droplet surface, which notably promoted polysaccharide molecules to attach to more than one droplet, resulting in the occurrence of bridging flocculation with large average particle sizes [14].

After 10 days, the filled hydrogels showed a stability phenomenon that without creaming from 0 to 300 mM (Fig. 2B). However, when the NaCl concentration further increased, the filled hydrogels showed stratification phenomenon that the upper layer was creamy and the layer phase was transparent continuous phase (Fig. 2B), indicating that droplets became aggregated and appeared phase separation phenomenon due to the imbalance between oil droplets and water phase. Moreover, this phenomenon be-came more and more obvious with increasing NaCl concentration from 400 to 500 mM. This indicated that the electrostatic repulsion between droplets decreased due to the electrostatic screening of NaCl, thereby decreasing the stability of filled hydrogels. In addition, the filled hydrogels stored for 10 days had higher particle sizes (D4,3 and D3,2) than those of the freshly prepared filled hydrogel (P < 0.05), indicating that the stability diminished with an increase in storage period. For filled hydrogels stored for 10 days, the droplet diameters of filled hydrogels increased and the large particle peak in the particle size distribution of filled hydrogels shifted toward the right with an increasing NaCl level (Fig. 2D and Table 1), which was in accordance with the results of freshly prepared filled hydrogels. NaCl addition promoted the flocculation of protein on the surface of droplets and change the structure of interface, causing a larger and non-uniform distribution of droplets. These consequences confirmed that the addition of NaCl might reduce the physical stability of filled hydrogels during storage.”

In addition, the initial data of zeta-potential measurements showed that there are no zeta-potential profiles, but has only a specific value for each sample. In our revised manuscript, we change the zeta-potential data in table to the zeta-potential diagrams.

We are indebted to the reviewer for this constructive suggestion to improve the quality of the manuscript. Thanks.

Q7: How about the effect of morphology of the hydrogel? Add SEM image.

A7: This is a good suggestion. According to the reviewer’s opinion, we have observed the surface morphology of filled hydrogels at difference NaCl concentrations via a cryo-scanning electron microscopy (Cryo-SEM). In our revised manuscript, we have added some statements about the surface morphology as follows:

In the section of “Materials and Methods

2.9 Cryo-scanning electron microscopy (Cryo-SEM)

Micromorphology of each sample was observed with scanned images magnified 10, 000 × using a Cryo-SEM equipped with a cryo-transfer system (S-3400N, Hitachi, Japan) ac-cording to the methods of our previous study [19].

In the section of “Results and discussion

3.7. Microscopic morphology

The microstructure of NaCl-treated filled hydrogels are shown in Fig. 7A. The droplets were coated by a molecular layer of whey proteins and polysaccharide, contributing to the stabilization of filled hydrogels. Wijaya et al. reported that the stabilization of emulsion stabilized by complexes was due to the intermixed layer of whey protein and low methoxyl pectin or the combination of an interpolyelectrolyte network that formed by pectin chains linked together with clustered protein [42]. With an increase of NaCl concentration, the lipid droplets size gradually increased, which was supported the results of particle size. Moreover, when the NaCl concentration increased, the surface of droplets had some irregular large aggregates. Taha et al. reported that the increase of electrostatic screening induced by NaCl would enhance the interparticle interaction of protein and increase the protein aggregation at the oil/water interface, thus resulting to the flocculation of droplets in emulsion [43]. Moreover, under relatively higher NaCl concentration (such as 300, 400 and 500 mM), some irregular large aggregates gradually increased in the continuous phase surrounding the droplets, especially indicated as Fig. 7E and 7F. The addition of NaCl may be led to the electrostatic screening and decreased the electrostatic attraction between the protein and pectin molecules, resulting in the protein molecules moved from the filled hydrogels into continuous phase [44]. Based on the above discussion, Cryo-SEM analysis suggested that the addition of NaCl could account for the physical stability of filled hydrogels.

The five newly cited references are as follows:

  1. Wijaya, W.; Meeren, P.V.; Wijaya, C.H.; Patel, A. R. High internal phase emulsions stabilized solely by whey protein iso-late-low methoxyl pectin complexes: effect of pH and polymer concentration. Food Funct. 2016, 8, 584-594. https://doi.org/10.1039/c6fo01027j
  2. Taha, A.; Ahmed, E.; Hu, T.; Xu, X.Y.; Pan, S.Y., Hu, H. Effects of different ionic strengths on the physicochemical properties of plant and animal proteins-stabilized emulsions fabricated using ultrasound emulsification. Ultrason. Sonochem. 2019, 58, 104627. https://doi.org/10.1016/j.ultsonch.2019.104627.
  3. Chung, C.; Degner, B.; Decker, E.A.; McClements, D.J. Oil-filled hydrogel particles for reduced-fat food applications: Fab-rication, characterization, and properties. Innov. Food Sci. Emerg. Technol. 2013, 20, 324-33. http://dx.doi.org/10.1016/j.ifset.2013.08.006

We are indebted to the reviewer for this constructive suggestion to improve the quality of the manuscript. Thanks.

Q8: Add apparent image of all the hydrogel.

A8: This is a good suggestion. According to the reviewer’s opinion, we have added the apparent image of filled hydrogels at difference NaCl concentration as follows:

“All freshly prepared filled hydrogels showed a uniform and white appearance in Fig 2A, which was supported the whiteness results in colour analysis. Moreover, the particle size distributions of freshly prepared filled hydrogels at difference NaCl concentration are shown in Fig. 2C. The large particle peak in the particle size distribution of filled hydrogels shifted toward the right with the increase of NaCl level increased from 0 to 500 mM, suggesting an increase of particle size.”

“After 10 days, the filled hydrogels showed a stability phenomenon that without creaming from 0 to 300 mM (Fig. 2B). However, when the NaCl concentration further increased, the filled hydrogels showed stratification phenomenon that the upper layer was creamy and the layer phase was transparent continuous phase (Fig. 2B), indicating that droplets became aggregated and appeared phase separation phenomenon due to the imbalance between oil droplets and water phase. Moreover, this phenomenon be-came more and more obvious with increasing NaCl concentration from 400 to 500 mM. This indicated that the electrostatic repulsion between droplets decreased due to the electrostatic screening of NaCl, thereby decreasing the stability of filled hydrogels. In addition, the filled hydrogels stored for 10 days had higher particle sizes (D4,3 and D3,2) than those of the freshly prepared filled hydrogel (P < 0.05), indicating that the stability diminished with an increase in storage period. For filled hydrogels stored for 10 days, the droplet diameters of filled hydrogels increased and the large particle peak in the particle size distribution of filled hydrogels shifted toward the right with an increasing NaCl level (Fig. 2D and Table 1), which was in accordance with the results of freshly prepared filled hydrogels.”

We are indebted to the reviewer for this constructive suggestion to improve the quality of the manuscript. Thanks.

Q9: Authors are advised to add the pictorial mechanistic aspect of NaCl on the lipid and protein oxidation and discuss it in the text.

A9: This is a good suggestion. According to the reviewer’s opinion, we have added the pictorial mechanistic of NaCl on the lipid and protein oxidation and discuss it in our revised manuscript as follows:

“Based on above results of lipids oxidation and protein oxidation, the effect of NaCl on lipids oxidation and protein oxidation of filled hydrogels are shown in Fig. 10. The addition of NaCl could increase the catalytic activity of iron and promote the dissociation of H atom of the adjacent methylene carbon atom of lipids double bond, thus leading to the formation of primary lipid oxidation products and secondary lipid oxidation products. Moreover, NaCl could reduce the negative charge of the filled hydrogel and weaken the electrostatic adsorption force between the positively charged metal ions, resulting in weakening the antioxidant capacity of protein and promoting the oxidation reaction of the droplets [46]. In addition, based on the microstructure analysis, the protein aggregation into continuous phase decreased the antioxidant capacity, such as the antioxidant capacity, thus the oxidation of unsaturated fatty acids under certain conditions is promoted. For the protein oxidation, NaCl addition led to the ex-tensive protein aggregation and increase steric hindrance, decreasing the ABTS+ free radical scavenging ability metal ion chelating ability and reducing ability, thus promoting the transform of tryptophan residues that located in the internal surface of the droplets form radicals and the formation of protein oxidation products. In addition, the addition of NaCl promoted the formation of primary lipid oxidation products and secondary lipid oxidation products, which may be contributed the reaction with tryptophan peroxyl radicals and accelerate the oxidation of the protein [49].”

We are indebted to the reviewer for this constructive suggestion to improve the quality of the manuscript. Thanks.

Q10: The discussion could be more solid. Revise it carefully.

A10: This is a good suggestion. According to the reviewer’s opinion, we have added some statements about the discussion as follows:

“NaCl addition promoted the flocculation of protein on the surface of droplets and change the structure of interface, causing a larger and non-uniform distribution of droplets. These consequences confirmed that the addition of NaCl might reduce the physical stability of filled hydrogels during storage.”

“Moreover, higher NaCl concentration may be produced the electrostatic shielding for protein and decreased the solubility of protein, which caused the inhibition of hydrogen bonds and hydrophobic interactions between protein and polysaccharide, leading to the lower G′ and higher tan δ [36].”

We are indebted to the reviewer for this constructive suggestion to improve the quality of the manuscript. Thanks.

Q11: Also, carefully revise the manuscript for linguistic and typos errors.

A11: This is a good suggestion. According to the reviewer’s opinion, we have revised the typos errors and invited a native speaker to help us revise the English grammar throughout the paper. We are indebted to the reviewer for pointing out this problem and we will pay more attention to these details in the future. Thanks.

Round 2

Reviewer 1 Report

Many thanks for the response. Very good research. 

Reviewer 2 Report

The revision has been made as per the reviewer's suggestions and hence the manuscript can be accepted for publication.